

**Refined classification and characterization of atmospheric new particle**
**formation events using air ions**
Lubna Dada[1*], Robert Chellapermal[1], Stephany Buenrostro Mazon[1], Pauli Paasonen[1], Janne Lampilahti[1],
Hanna E. Manninen[1,2], Heikki Junninen[1,3], Tuukka Petäjä[1,4], Veli-Matti Kerminen[1], and Markku
Kulmala[1,4,5]
[1] Institute for Atmospheric and Earth System Research, University of Helsinki, Helsinki, Finland
[2] Experimental Physics Department, CERN, 1211 Geneva, Switzerland
[3] Institute of Physics, University of Tartu, Ülikooli 18, EE-50090 Tartu, Estonia
[4] Aerosol and Haze Laboratory, Beijing Advanced Innovation Center for Soft Matter Science and Engineering, Beijing
University of Chemical Technology, Beijing, China
[5] Joint International Research Laboratory of Atmospheric and Earth System Sciences, Nanjing University, Nanjing, China
*Correspondence to*: Lubna Dada (lubna.dada@helsinki.fi)
**Abstract.** Atmospheric new particle formation (NPF) is a world-wide observed phenomenon that affects the human health
and the global climate. With the growing network of global atmospheric measurement stations, efforts towards investigating
NPF have increased. In this study, we present an automated method to classify days into four categories including NPF events,
non-events and two classes in between, which then ensures the reproducibility and minimizes the man-hours spent on manual
classification. We applied our automated method to 10 years of data collected at the SMEAR II measurement station in
Hyytiälä, southern Finland. In contrast to the traditionally-applied classification methods which categorize days into events,
non-events and ambiguous days as undefined days, our method is able to classify the undefined days as it accesses the initial
steps of NPF at sub-3 nm sizes. Our results show that on ~24% of the days in Hyytiälä, a regional NPF event occurred and
was characterized by a 'nice weather' and favorable conditions such as a clear sky and low condensation sink. Another class
found in Hyytiälä is the transported event class, which seems to be NPF carried horizontally or vertically to our measurement
location and it occurred on 17% of the total studied days. Additionally, we found that an ion burst, where the ions apparently
fail to grow to larger sizes, occurred on 18% of the days in Hyytiälä. The transported events and ion bursts were characterized
by less favorable ambient conditions than regional NPF events, and thus experienced interrupted particle formation or growth.
Non-events occurred on 41 % of the days and were characterized by a complete cloud cover and high relative humidity.
Moreover, for the regional NPF events occurring at the measurement site, the method identifies the start time, peak time and
end time, which helps us focus on variables within an exact time window to better understand NPF in a process level. Our
automated method can be modified to work in other measurement locations where NPF is observed.
Keywords: NPF events, air ions, intermediate ions, boreal forest






## 1    Introduction

New particle formation (NPF) is an atmospheric phenomenon that results in a big addition to aerosol load in the global troposphere Spracklen et al. (2010);Kerminen V-M. (2018) . NPF is observed frequently in different environments around the globe, ranging from pristine locations (Siberia –Kulmala et al. (2011);Asmi et al. (2016)), to boreal forests (Hyytiälä -Kulmala et al. (2013);Nieminen et al. (2014)), tropical forests (Amazon -Artaxo et al. (2013);Wimmer et al. (2017)), mountain tops (Jungfraujoch–Bianchi et al. (2016)), semi-polluted cities (European cities -Manninen et al. (2010) ) and even heavily polluted mega cities (China -Kulmala et al. (2016);(2017);Wang et al. (2017). The freshly formed particles that grow to larger sizes contribute largely to the cloud condensation nuclei load in the atmosphere (Merikanto et al., 2009; Kerminen et al., 2012;Salma et al., 2016) and thus indirectly affect the climate.

In order to comprehend the phenomenon of NPF in a specific location, we first need to understand its frequency and characteristics as well as particle formation and growth rates associated with it. With the growing number of global stations (Kulmala, 2018), an automatic method is needed to classify the days into events and non-events. In addition to minimizing the effort of manual event classification, an automated method tends also to reduce any human error. In this study, we present an automated method which classifies days into four classes according to the observed characteristics of 2-4 nm sized air ions and 7-25 nm sized particles. The original classification method of days as events, non-events and undefined days was proposed by (Dal Maso et al., 2005), and later modified by (Kulmala et al., 2012), and is based on particle measurements starting from about 3 nm in particle mobility diameter, thus missing the initial steps of NPF. With the increased development of instrumentation, we are able to access sub-3 nm clusters and refine our classification method to account for the very initial steps of NPF. The classification proposed here divides days into regional events, transported events, ion bursts and non-events, thus excluding any 'undefined' days, which minimizes the number of days usually excluded from further data analysis. Furthermore, our automated method identifies the start, peak and end time of daytime regional events or ion bursts. By identifying the start and end times, we are able to concentrate on the conditions present during the actual NPF time window.

Our study focuses on the NPF occurring in Hyytiälä, a boreal forest site in southern Finland where the SMEAR II (Station for Measuring Forest Ecosystem-Atmosphere Relations) measurement station is located (Hari and Kulmala, 2005). The dataset collected at the station sums up more than 22 years of particle, meteorological and gas data, making extensive analyses of NPF and related parameters possible. Besides studying NPF occurrence in Hyytiälä, our method can be applied to other locations where NPF is observed, enabling scientists studying particle formation to focus on specific time windows by which active NPF occurs. Our specific aims in this study are i) to automatically classify days in Hyytiälä according to their initial NPF steps, ii) to minimize the number of undefined days by refining the classification, iii) to investigate different characteristics of classified days, iv) to identify the start, peak and end times of regional events and, thereby, v) to create a time series which allows us to focus on the exact time period during which a regional new particle formation event has occurred.

65



## 2   Materials and Methods

### 2.1   Measurement location

The main results of our study are based on the measurements collected at the SMEAR II) station located in the boreal forest site in Hyytiälä, Southern Finland (61°51'N, 24°17'E, 181 m a.s.l). The station has accumulated 22 years of comprehensive measurements including particle, radiation, gas, meteorological and complementary data. The location is considered a semi-clean boreal forest environment as it is far from anthropogenic pollutants (Asmi et al., 2011) and thus represents the northern-hemisphere boreal forests. A more detailed description of the site and the ongoing measurements can be found in Hari and Kulmala (2005) and Nieminen et al. (2014).

### 2.2   Instrumentation

The traditional classification of days as NPF events and non-events follows the method proposed by Dal Maso et al. (2005); Kulmala et al. (2012). For this classification method, the particle number-size distributions measured with a twin-DMPS (Differential Mobility Particle Sizer) system(Aalto et al., 2001), were used. The twin DMPS system measured the aerosol number-size distribution over the size range 3-500 nm until 2004 and over the size range 3-1000 nm from 2005 onwards. The DMPS measurements are also used to calculate the condensation sink (CS) which is the rate at which non-volatile vapors condense onto a pre-existing particles (Kulmala et al., 2012).

For our proposed automated classification method, the mobility distributions of charged and neutral aerosol particles and clusters in the size range of 0.8–47 nm and 2–42 nm, respectively, were measured with a Neutral cluster and Air Ion Spectrometer (NAIS, Airel Ltd., Estonia, (Manninen et al., 2016;Manninen et al., 2009;Mirme and Mirme, 2013) between 2006 and 2015. No measurements using the NAIS were made during year 2008 when the instrument was used for an intensive campaign. Particle and air ion data are available in two-minute time steps.

The air temperature and the relative humidity are measured with 4-wired PT-100 sensors and relative humidity sensors (Rotronic Hygromet MP102H with Hygroclip HC2-S3, Rotronic AG, Bassersdorf, Switzerland) on a mast at a height level of 16.8 m, respectively. The temperature and relative humidity data are provided as 30-minute averages. Solar radiation in the wavelengths of global radiation (0.30-4.8 µm) is monitored using pyranometers (SL 501A UVB, Solar Light, Philadelphia, PA, USA; Reeman TP 3, Astrodata, Tõravere, Tartumaa, Estonia until June 2008, and Middleton Solar SK08, Middleton Solar, Yarraville, Australia since June 2008) above the forest at 18 m. We used global radiation data for calculating the cloudiness parameter ($P$), which is the ratio of global radiation to theoretical maximum radiation arriving at Hyytiälä, by following the method proposed by Dada et al. (2017). Values of $P \leq 0.3$ represent a complete cloud cover while values of $P \geq 0.8$ can be considered to represent clear-sky conditions.

### 2.3   Event classification decision tree

Based on the concentrations of 2 – 4 nm ions, we are able to detect the initial steps of cluster formation (see Leino et al. (2016)),which would not be possible using the DMPS system alone and the traditional classification. This small size window available from the NAIS operating in ion mode gives an additional opportunity to investigate sub-3 nm clusters. Accordingly, we are able to estimate whether a regional NPF event occurred within the air mass in which the observations were made, or elsewhere and then carried to our measurement location. Similarly, undefined days are identified based on their sub-3 nm characteristics. We present in Figure 1 our refined classification decision tree and apply it to Hyytiälä data in this study. In order to attain this classification, we rely on the initial steps of cluster formation and their further growth, which we monitor using an automatic method. Since in our study we are interested in daytime NPF, we chose the time window between 06:00 and 19:00 when monitoring aerosol number concentrations. However, the automated method can be tweaked to include evening or night time event classification in places where these event types are present.

Our decision tree (Figure 1) first examines 2–4 nm ion concentrations representing the initial step of new particle formation. A notable increase in their concentration is interpreted as ion clustering on site. To be accounted as an increase, the number concentration of ions after 06:00 must increase above a relative threshold and persist for more than 1 hour. This threshold is calculated from ion concentration averaged over the time period 00:00–04:00 multiplied by a scaling factor (Figure 2A); we chose this time window as background as it is outside the time window when night time ion clusters are observed (Buenrostro Mazon et al., 2016;Rose et al., 2018). To be accounted as a notable increase past the threshold value, a concentration of 20



ions/cm³ should be reached and should last for at least 1 hour. We chose the aforementioned value as it has been found to be
an indicator for NPF in Hyytiälä (Leino et al., 2016). If this criterion is met, these ions are expected to either grow into bigger
sizes and lead to regional NPF events (RE), or fail to grow further, in which case the event are identified as ion bursts (IB)
that do not form new particles.
To decide whether the particle growth is observed, particle concentrations in the size range of $7 - 25$ nm are examined. These
particles represent the growth phase of freshly-formed clusters. Since in Hyytiälä growth rates of $4 - 7$ nm particles is reported
to lie between 0.8 and 17 nm/h (Average 3.8 nm/h ) (Yli-Juuti et al., 2011), we considered a time delay of 1 to 8 hours between
the initial increase of ion ($2 - 4$ nm) concentrations and particle ($7 - 25$ nm) concentrations. To be considered as an increase,
the particle number concentration should exceed a relative threshold which in this case is the number concentration averaged
over the time period of 03:00–05:00 (Figure 2B). We determined the background time window by comparing the automatic
method to a manual classification that we performed for the years 2013-2014 from our data set. The increase in concentration
should last for ~1.5 hr (100 minutes) and reach a peak of at least 3000 particles/cm³. On one hand, if both 2 - 4 nm ions and
$7 - 25$ nm particles are present, the time period is considered as a regional event (RE). On the other hand, if the 2 - 4 nm ions
are present but they do not grow to form $7 - 25$ nm particles, the time period is classified as an ion burst (IB).  Moreover, if 2
$- 4$ nm ions are not present, but we observe an increase in the particles, this leads to the assumption that the NPF event did
not occur at the measurement location but was carried horizontally or vertically to our site (Leino et al., 2018). The latter has
been previously described as a tail event (Buenrostro Mazon et al., 2009) or a transported event (TE). However, if neither
criterion is met, which means that neither $2 - 4$ nm ions nor $7 - 25$ nm particles are present in sufficient concentrations, the
time period is then classified as a non-event (NE).
*2.4    Description of the automated method*
Our automatic method selects the start time, peak time and end time of negative NAIS ions in the size range $2 - 4$ nm. The
growth to an event is confirmed by an accompanying peak in the 7 - 25 nm particles measured by the NAIS. The outcome of
the automatic method is the classification of days into the four classes, as well as a time series that identifies the time period
of regional events and ion bursts in Hyytiälä (Pathways RE and IB in Figure 1).
First, to investigate the appearance of $2 - 4$ nm ions, the precipitation time stamps are excluded from our analysis as they
interfere with the ion data (Leino et al., 2016), resulting in misinterpretations. After that, the ion concentrations are smoothed
using Savitsky-Golay filter (DeSerio, 2008). We then search for an increase in the ion concentration that lasts for 12
consecutive points (5 minutes each) above a threshold value and reaches values greater than 20 cm⁻³ (Leino et al., 2016). A
maximum of 3 drops below the threshold value are allowed (Figure 2A). Finally, the method looks for a peak in the 7 - 25 nm
particle concentration to identify the appearance of a growth phase (Figure 2B). The peak requires 15 consecutive points (5
minutes each) having concentrations larger than the threshold value and reaching a value larger than 3000 cm⁻³. Also, a
maximum of 3 drops below the threshold value are allowed. Accordingly, each time stamp is classified.
*2.5    Start time, peak time and end time determination*
The start time, peak times and end times for regional events and ion bursts are defined based on the $2 - 4$ nm ion concentration
as follows: i) The start time is the first crossing of the threshold line which lasts for more than 12 consecutive points, ii) the
peak time is when the concentration reaches the maximum and iii) the end time is the first trough after crossing the threshold
line into lower concentrations which remains below the threshold for more than 3 consecutive points. An example day is
demonstrated in Figure 2A. The threshold is taken as the $2 - 4$ nm ion concentration averaged over the time period 00:00–
04:00 multiplied by a scaling factor of 7. Our scaling factor was determined after we did a comparison with the manual
classification of the data for the years 2013-2014.
**3    Results and Discussion**
*3.1    Event Classification*
Our classification categorizes the days in Hyytiälä into four different categories following the pathway chart in Figure 1. Type
RE, or regional NPF events, are those which are initiated over a large area including the measurement location and the particles
continue to grow to bigger sizes. The type TE, or transported events (also known as tail events by Buenrostro Mazon et al.
(2009)), are events whose beginning is not detected as it does not occur at the immediate vicinity of our measurement site.





Such events could be attributed to events that were initiated outside our measurement site and transported to Hyytiälä (Leino
et al. 2018). The aforementioned hypotheses could explain the observation that TE typically occur at around midday or later
in the afternoon, while RE tend to occur concurrent with sunrise. The type IB, or ion bursts, are attempts of NPF, during
which clusters form in Hyytiälä, however, they do not grow beyond a few nanometers in diameter. Changes in atmospheric
conditions that could cause the limited, or interrupted, growth of the clusters are assessed in more detail in section 3.3. Finally,
non-events (NE) are days for which we do not observe a forming mode of 2 – 4 nm ions nor a growing mode of 7 - 25 particles.
*3.2    Frequency of Events*
For 10 years of data (2006 – 2016), excluding the days with missing NAIS data when the instrument was under maintenance
or on campaigns, we classified a total of 2134 days. Using our refined classification method, we were able to classify the days
into 4 categories as follows (Figure 3): 551 RE (24%), 410 TE (18%), 415 IB (18%) and 938 NE (40%). This refined
classification is able to classify all days into categories and thus eliminate the undefined days that usually constitute around
40% of all the days in our location (Dal Maso et al., 2005;Buenrostro Mazon et al., 2009).
Moreover, we studied the inter-annual variation of each of the classes (Figure 4A). In general, RE constitute 20-30% of the
total classified days. In 2006, the measurement started in September, which explains a lower fraction of RE. The gap in the
analysis in 2008 is explained by a campaign during which the NAIS data is not available (Manninen et al., 2010). The data in
2009 includes data from spring only, which explains the high frequency of RE in 2009. While we can observe changes in the
frequency of RE between the years, no clear trend exists. The annual variation of TE follows that of RE, also with no specific
trend over the years. The type IB appears to have an almost constant fraction over the years. Finally, NE constitute between
40 and 50% of the days, except in 2009 which has the bias for spring favoring RE.
The monthly variation of RE follows the typical yearly cycle of NPF, with a peak in spring, followed by a smaller peak in
autumn(Dal Maso et al., 2005;Nieminen et al., 2014;Dada et al., 2017). Interestingly, the refined classification shows that the
events occurring in spring are mostly RE while those in autumn are dominated by TE. Additionally, RE rarely occur in winter,
appearing on less than 5% of the days. IB have a steady 10-20% occurrence during the year. Finally, NE occur on 60 to 70%
of winter days and less than 30% during spring. Interestingly, while previously it was understood that summer is dominated
by NE (Nieminen et al., 2014;Dada et al., 2017), the refined classification shows that both TE and IB are frequent during
summer, complementing observations by Buenrostro Mazon et al. (2011) who reported 'failed events' during summer.

*3.3    Characteristics of RE, TE, IB and NE*
For a regional event to take place, favorable conditions need to be present. These include a low condensation sink, low relative
humidity, moderate temperature and plenty of radiation available during a clear sky (Dada et al., 2017;Hyvönen et al.,
2005;Nieminen et al., 2014;Nieminen et al., 2015). In Figure 5, we present the characteristics of each type of event classified
in terms of Condensation Sink (CS), relative humidity (RH), Temperature ($T$) and Cloudiness ($P$). The data in the plots
represent half-hour averages of each variable between 7:00 and 12:00 during spring (March – May). We chose this season in
order to capture the maximum NPF events and this time window in order to be consistent between all four studied classes. As
expected, the median CS observed on RE was $1.7 \times 10^{-3}\,s^{-1}$ which is a factor of 2 lower than CS observed on TE days or on
NE days ($3 \times 10^{-3}\,s^{-1}$). To our understanding, high CS inhibits NPF, so that its higher values during the days classified as TE
forbid the initial formation of particles at the measurement site. IB, on the other hand, are potential regional events whose
growth has been interrupted. Since the median CS during IB was not high ($2.5 \times 10^{-3}\,s^{-1}$), it does not explain the discontinuous
growth of the clusters during these events. We proceed to study the effect of $T$ on the occurrence of each class of events. Since
the data in Figure 5 are measurements during spring, the median value of temperature (2-7 °C) was rather similar on all days
and no specific trend or exception could be found.
In addition to CS and $T$, RH and cloudiness ($P$) play an important role in the occurrence of NPF (Dada et al., 2017;Hamed et
al., 2011). A regional NPF event is more likely to occur on a clear-sky day rather than on a cloudy day. This conclusion is
demonstrated nicely in Figure 5 which shows that the median value of $P$ was close to 0.8 on the RE days and closer to 0.3 on
NE day. TE usually took place when the conditions within the boundary layer were not favorable for a regional NPF to occur.
However, the particle growth was much less sensitive to environmental conditions: a particle growth was often observed
during all times of day and in every season, also on days (and nights) when NPF did not take place (Paasonen et al., 2018).



Combined with a higher CS, the value of *P* was much lower on TE days than on RE days, describing a semi-cloudy day
unfavorable for NPF to occur within the boundary layer, which could result in the occurrence of a TE in locations where the
conditions are conducive enough to NPF. It is, however, important to mention that it is possible to have a regional NPF episode
taking place simultaneously with a transported one, and when the latter is transported it gets mixed with the regional NPF so
that this situation will be classified as a RE. Finally, since ion bursts are attempts of an event but do not grow, an interrupted
clear sky could explain this phenomenon: for instance a sudden appearance of a cloud would result in the interruption of NPF
(Baranizadeh et al., 2014), which then remains as an ion burst only. Finally, the RH, which in general correlates with
cloudiness, showed a nice pattern between the event classes: RH was the lowest for RE and the highest for NE, and it fairly
reflects cloudiness.
*3.4   Start times, peak time and end time of RE*
Our method makes it possible to detect the start, peak and end times of every regional event classified during our study period.
Although several previous studies state that that the occurrence of NPF starts with sunrise and peaks around midday, very few
investigations have considered occurrence times accurately. We derived the start, peak and end times from 2 - 4 nm ions
automatically, as mentioned in sections 2.4 and 2.5. Our results (Figure 6) show that indeed RE tended occur after sunrise and
prior to noon, with the maximum number of days occurring between sunrise and 5 hours past sunrise. The peak times of the
events had the most frequent occurrence at 5 to 6 hours after sunrise, which is between 10:30 and 11:30 local time,
complementing our previous assumption that NPF peaks before noon. Finally, the ending times of the events had the most
frequent occurrence at 10 hours after sunrise. The importance of the identification of the exact start and end times of the
process helps to increase our understanding on the processes governing the NPF phenomenon. More specifically, they allow
forming a time series where NPF is separated from non-event times, making it possible to compare the parameters responsible
for the NPF process within appropriate time frames.
*3.5   Comparison to previous classification*
In order to estimate the goodness of our automatic method, it is crucial to compare our results with the previous classifications
(Dal Maso et al., 2005;Kulmala et al., 2012). Although such a comparison is not straightforward, we show one version of
such a comparison in Figure 7. On the x-axis, the original classified days are shown, and the refined classes are shown on the
y-axis as a fraction of each original class. For example, 65% of the originally-classified event days (event days make 25% of
the total days in Hyytiälä according to the original classification) were found to be RE, 10% were TE and 14% were IB. The
remaining 11% were considered as misclassified or bad data (by manual classification) and were excluded from the plot. In
total, our automatic method was able to classify 89% of the original NPF events into some of the new event classes (RE, TE
or IB). The original non-events (which made 40% of the total days) were split between the TE (20%), IB (19%) and NE
(53%). The remaining 8% were bad data according to the manual classification and were excluded from the plot.
Finally, undefined days, which according to the traditional classification were 35% of the total days, were split between all
the classes. Our results show that 17% of those were RE, 21% were TE, 19% were IB and 42% were non-events. Those days
were usually excluded from further analysis because they did not belong to a defined class according to the original
classification method. Previous extensive studies of undefined days in Hyytiälä by Buenrostro Mazon et al. (2009) showed
that a fraction of undefined days resembles interrupted events which, in our case, were 83% of the days (TE, IB or NE), and
which all in all were related to unfavorable conditions for regional NPF. The interruption mechanisms may include appearance
of clouds (Dada et al., 2017), resulting in decreased radiation essential for particle formation and growth (Jokinen et al., 2017).
Others include a change of arriving air masses from a clean to a rather polluted sector (Sogacheva et al., 2005). Also, the
growth can be interrupted by a sudden appearance of a cloud (Baranizadeh et al., 2014;Dada et al., 2017).
**4   Conclusions**
Using 10 years of measurement using the NAIS at SMEAR II station, we were able to create an automated method to classify
days into 4 classes based on their ion (2 – 4 nm) and particle (7 - 25 nm) number concentrations, including regional events,
transported events, ion bursts and non-events. Our method minimizes the efforts used in manual day-by-day classification as
well as the errors due to human bias. In addition, our method allows for the complete classification (sub-3 nm) of all days, i.e.
reduces the number of previously known 'undefined days', which have always been excluded from previous analyses.



Our results show that on ~ 40% of the days during spring in Hyytiälä, a regional NPF event occurs and is characterized by a
set of favorable conditions, such as a clear sky, low condensation sink, medium temperature and low relative humidity. On
the contrary, NE were ~25 % of the days and were characterized by a complete cloud cover, high RH and high CS.
Interestingly, TE and IB fall in the category between RE and NE in this respect. While IB are interrupted growth of initially
started RE due to a probable change to polluted air mass or an appearance of a cloud, TE occurred on days when there was
little chance for the cluster to form within our measurement location but still they had a chance to grow if reaching our site.
Both IB and TE were characterized by intermediate values of CS, RH and $P$ compared with RE and NE. Moreover, using the
new method we are able to identify the start time, peak time and end time of events occurring in Hyytiälä. Our results show
that most RE started within 5 hours from the sunrise, peaked before noon, and ended 10 hours after sunrise. Finally, with
small changes the classification method can be applied to other places around the globe where NPF takes place providing
deeper understanding yet less effort for atmospheric scientists.




*Acknowledgements:* Lubna Dada acknowledges the doctoral programme in Atmospheric Sciences (ATM-DP, University of
Helsinki) for financial support. This project has received funding from the European Union's Horizon 2020 research. This
work was supported by the European Commission via projects ACTRIS2, European Research Council via ATM-GTP
(742206), and Academy of Finland Centre of Excellence in Atmospheric Sciences (grant number: 272041).
*Data availability:* Data measured at the SMEAR II station are available on the webpage: http://avaa.tdata.fi/web/smart/. The
classification, start times, peak times and end times are available from Lubna Dada (lubna.dada@helsinki.fi) upon request.

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




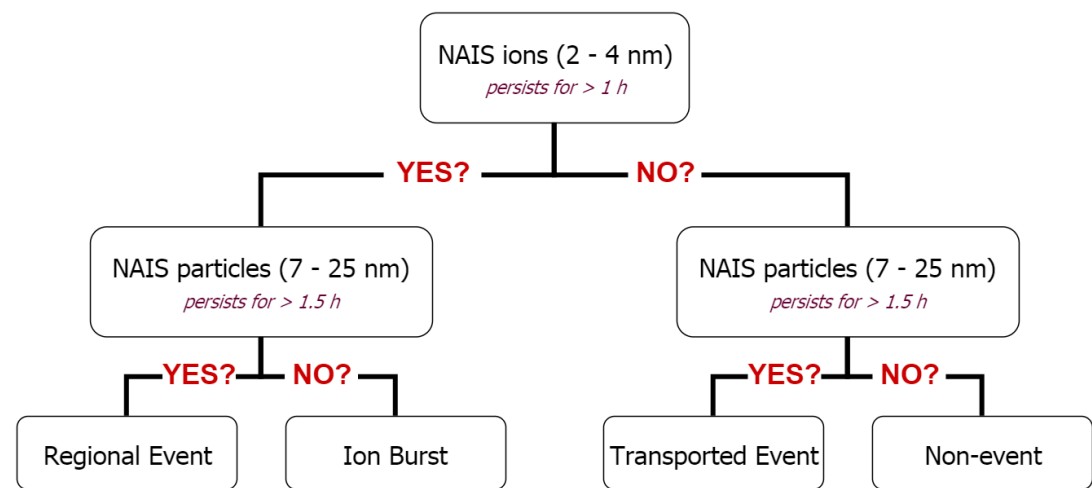


*Figure 1 A flow chart for the decision path during event classification in Hyytiälä using new classification method.*



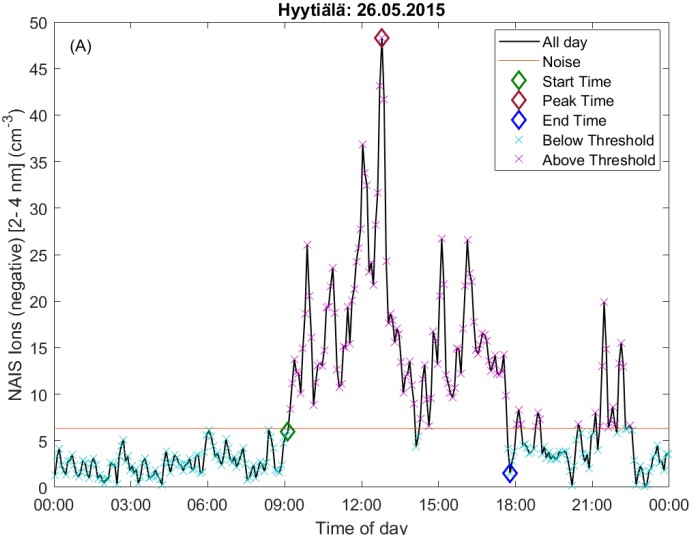

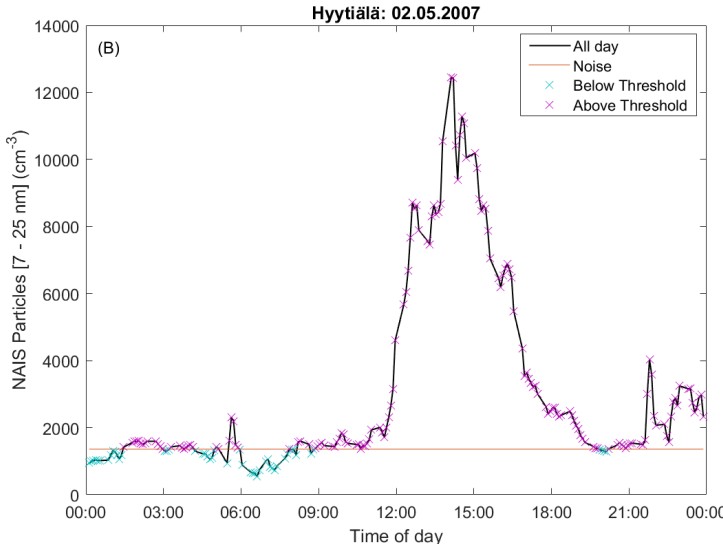

*Figure 2 Automatic method applied to (A) 2 – 4 nm ions (negative) example, ion concentration passed threshold and persisted > 1 hour and (B) 7 – 25 nm particles example, particle concentration passed threshold and persisted for > 1.5 hours.*



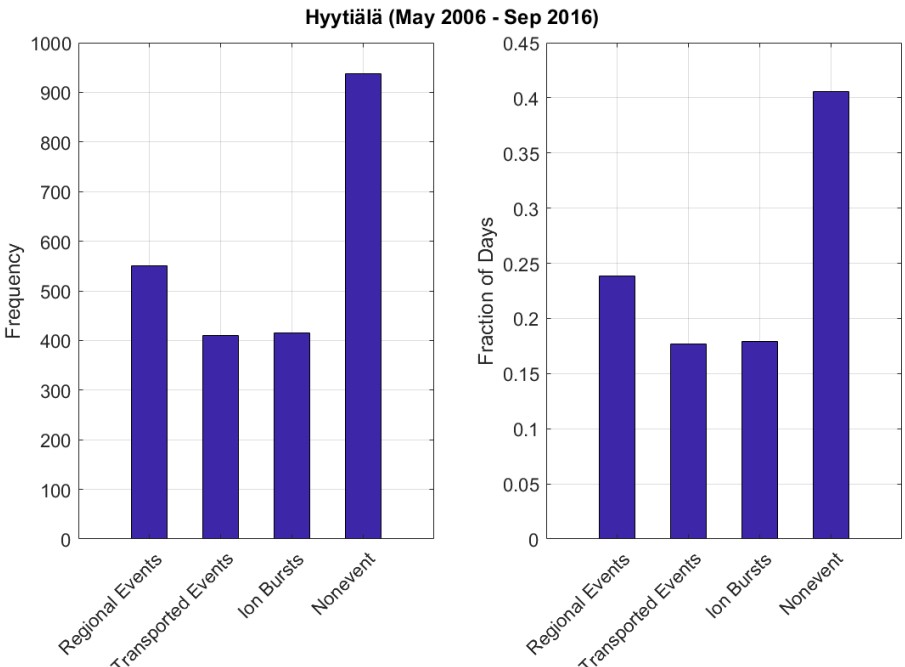


*Figure 3 Frequency and fraction of events, ions burst and non-events in Hyytiälä using the new classification method.*

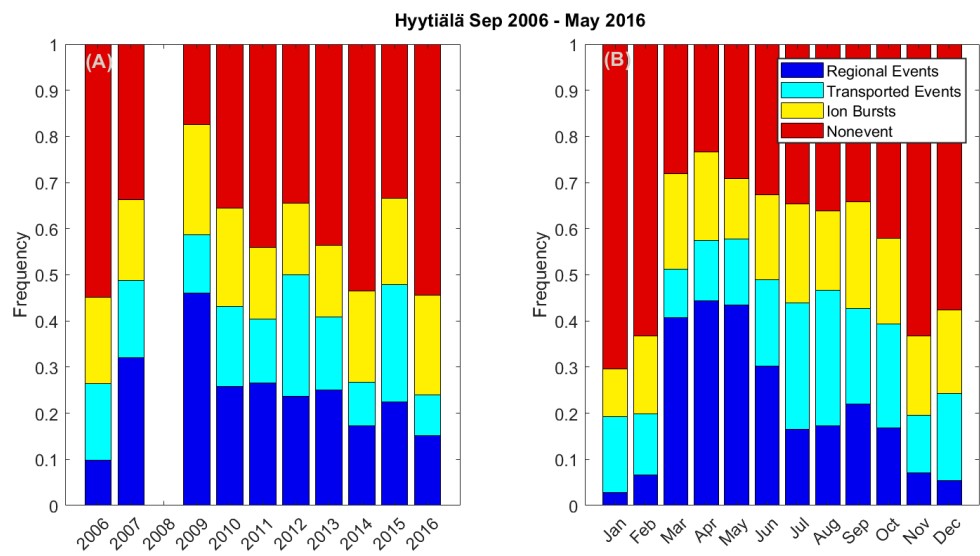


*Figure 4 (A) Yearly and (B) monthly fraction of days classified as Regional events (RE), Transported events (TE), Ion*
*bursts (IB), and non-events(NE) using the new classification method. The data of year 2009 is bias to spring months,*
*which could explain the much higher number of events. No data was available during 2008.*



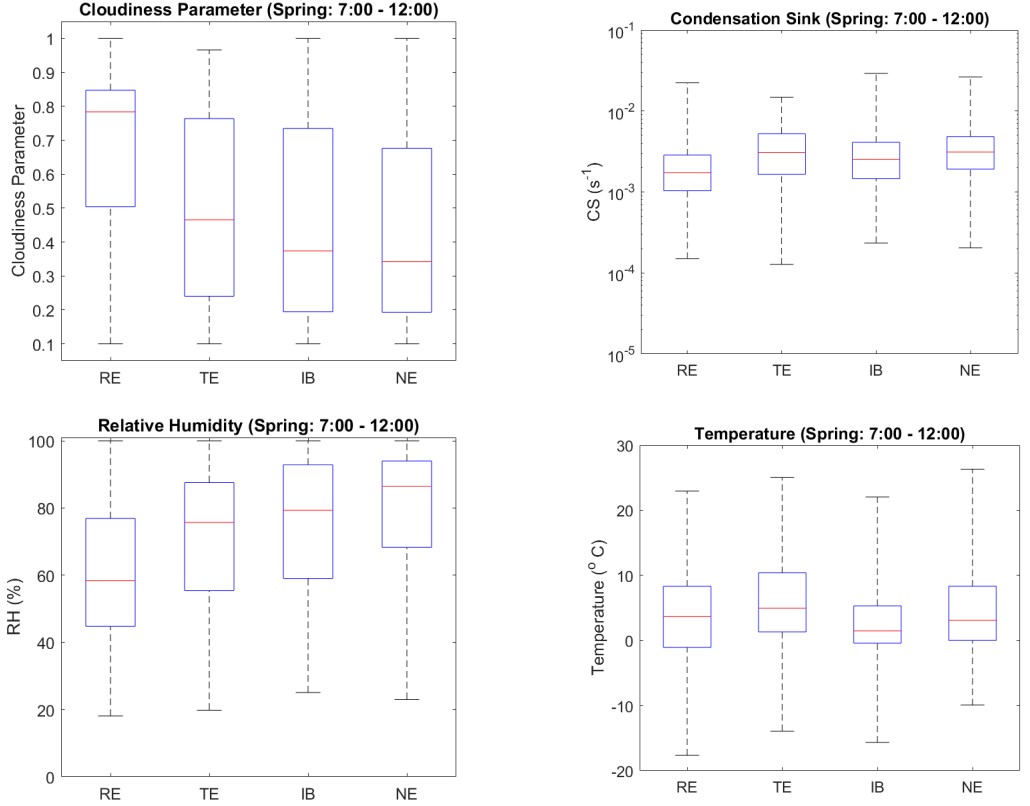

*Figure 5 (A) Cloudiness parameter, (B) condensation sink, (C) Relative humidity and (D) Temperature during different days classified with the new classification method for Spring (Mar-May) of 2006-2016 during maximum NPF window (7:00 – 12:00). The acronyms RE, TE, IB and NE stand for regional events, transported events, ions bursts and non-events, respectively. The red line represents the median of the data and the lower and upper edges of the box represent 25th and 75th percentiles of the data respectively. The lines extending from the central box represent the minimum and the maximum of the data inclusive.*






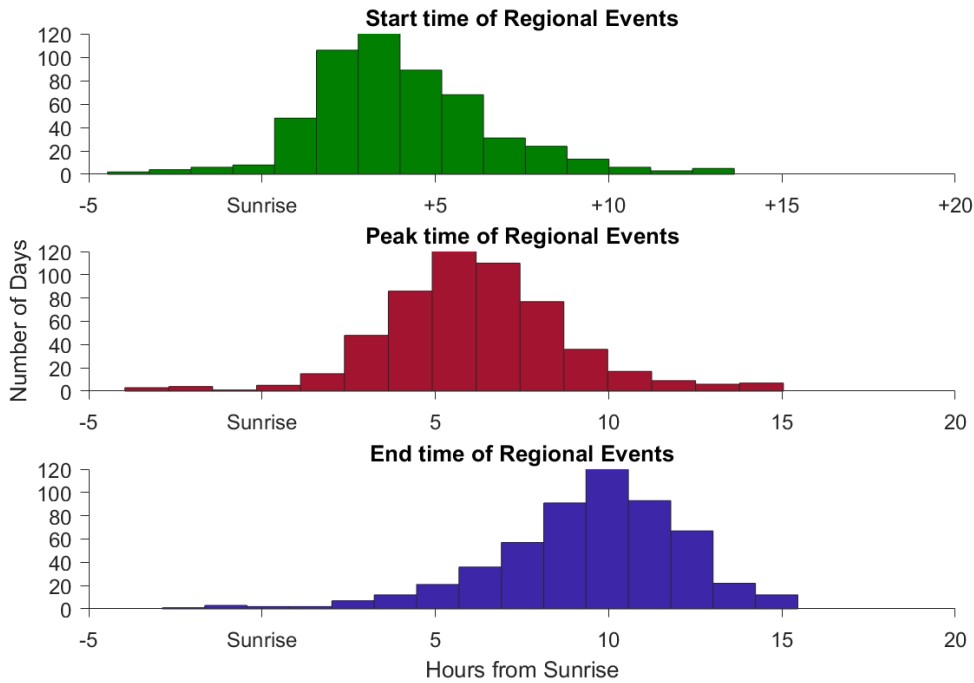


**Figure 6 Frequency of days at which regional events start, peak and end past sunrise. For example, most events start within 5 hours from sunrise.**



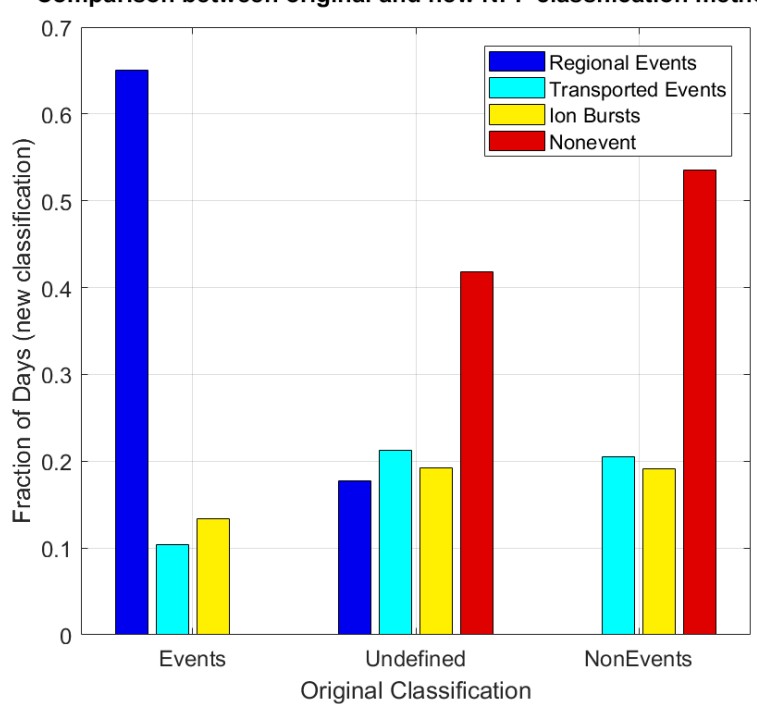


*Figure 7 Comparison between original and new NPF classification methods. The refined classification matches 94% with original event and non-event classification.*

