# Peer review of "Refined classification and characterization of atmospheric new particle formation events using air ions"

_Atmospheric Chemistry and Physics, 2018_

## Referee Comment (RC1) · Anonymous Referee #1 · 18 Sep 2018

This paper is an improvement of the Dal Maso et al. (2005) classification of New Particle Formation (NPF) days. That classification was limited to particles about 3 nm in mobility diameter and gave three classes: events, non-events and undefined days. In this paper a new proposal based on the ions participating in the nucleation is given and four classes can be obtained, excluding the undefined days. Other improvements have been incorporated, like the identification of regional and transported events. It has been applied to a large database and compared with the traditional manual procedure obtaining good results.

My main concern is that it uses NAIS data, an instrument not very spread in the com-

munity. This could limit the application of this new methodology. Anyway, this new method to classify NPF could be very useful in the near future. The results are discussed in an appropriate and balance way and the paper is well-structured. It is a significant work, concise and clearly written. I recommend publication in ACP and include some comments.

Specific comments:

Abstract: please, include the instrument used to obtain the data.

Line 69: "The station has accumulated 22 years..." Although the station is 22 years old, the dataset used in this paper is shorter, 11 years, please, indicate this here.

Lines 116-117: "To decide whether the particle growth is observed, particle concentrations in the size range of 7 – 25 nm are examined. These particles represent the growth phase of freshly-formed clusters." Is not there any other possibility? For example, could they come from bigger particles that have suffered shrinkage? It has been observed in some sites particles below 20 nm after shrinkage.

Point 2.4: In order to evaluate the improvement reached with this automated method, how long time do you need to classify a year using your method? And using the manual one?

Lines 219-222: "The peak times of the events had the most frequent occurrence at 5 to 6 hours after sunrise, which is between 10:30 and 11:30 local time, complementing our previous assumption that NPF peaks before noon. Finally, the ending times of the events had the most frequent occurrence at 10 hours after sunrise." Do these times depend on the season? Is this possible dependency seen in the data spread shown in the figure 6?

Lines 230-231: "For example, 65% of the originally-classified event days . . . were found to be RE, 10% were TE and 14% were IB". I don't understand how an event day obtained by the traditional method (Dal Maso et al. 2005) can be an ion burst event with

the new methodology. These IB events are characterized by not particle above 7 nm, so they should be considered non-event day by Dal Maso. Is there any reason for this?

Minor comments:

Check how the references are introduced in the text, sometimes not clear enough. Examples can be found in lines 35 and 48.

Line 40: Merikanto et al., 2009 is not included in the reference list

Line 41: Idem for Salma et al., 2016

Line 111: Idem for Rose et a., 2018

Line 118: Idem for Yli-Juuti et al., 2011

Lines 81-82: It is "the mobility distributions of charged and neutral aerosol particles and clusters in the size range of 0.8–47 nm and 2–42 nm, respectively, were measured with a Neutral cluster and Air Ion Spectrometer". I think it should be "the mobility distributions of neutral and charged aerosol particles and clusters..." to correlate with instrument list.

Line 165: "For 10 years of data (2006 – 2016)..." Previously it has been said from 2006-2015, I think the right period is until 2016, please, unify the dates.

Lines 243-244: "Also, the growth can be interrupted by a sudden appearance of a cloud (Baranizadeh et al., 2014;Dada et al., 2017)." This idea already appears two lines before. Remove one of the sentences

Line 302: the DeSerio (2008) reference is incomplete

———————————————————

---

## Referee Comment (RC2) · Anonymous Referee #2 · 8 Oct 2018

New particle formation has been demonstrated to play important roles in air quality and climate change. It's essential to classify the new particle formation events and non-events days accurately that can reduce the uncertainty when evaluating the contribution of NPF to aerosol and CCN budget. Previous methods were kind of subjective, and resulted in a poor comparability. This study present an automated method, which is more objective, to classify days into four categories including NPF events, non-events and two classes in between. This automated method was applied in a 10-year NAIS dataset at SMEAR II station. The classification using this methods almost matched the original method, but provided more reliable categories. Therefore, this automated method has the potential to be promoted widely. The manuscript is overall well written

and documented. The topic fits well in the scope of ACP. I recommend this manuscript can be published after some revisions.

Comments

1. A NAIS is needed to use this "new" method, which is not easy to be promoted. Can it be used with a SMPS or a DMPS? Hyde have SMPS/DMPS dataset, did the author compare the results that using a NAIS with a SMPS/DMPS? Are they identical?

2. Line 154-156: definition of region events is "initiated over a large area including the measurement location and the particles continue to grow to bigger sizes". Since the SMEAR II station is a surface measurement site, how did the author make sure the identified "regional event" occurred over a large area?

3. Transport events, is there any more evidence to support the definition? Any other possibility that other sources but not NPF contribute to the 7-25 nm particle?

4. Nighttime events: there are some regional events those were started and stopped before sunrise (Fig. 6)? Is it mean they are typical nighttime NPF events? Did they have the "banana" shape? If not, it means these events were not class A event, but still be defined as regional events (see comment 2)?

5. Figure 2: it's better to give an example to show the variation of 2-4 nm particles and 7-25 nm particles in one event.

---

## Author Comment (AC1) · 19 Nov 2018

Reply to Anonymous Referee #1

**This paper is an improvement of the Dal Maso et al. (2005) classification of New Particle Formation (NPF) days. That classification was limited to particles about 3 nm in mobility diameter and gave three classes: events, non-events and undefined days. In this paper a new proposal based on the ions participating in the nucleation is given and four classes can be obtained, excluding the undefined days. Other improvements have been incorporated, like the identification of regional and transported events. It has been applied to a large database and compared with the traditional manual procedure obtaining good results. My main concern is that it uses NAIS data, an instrument not very spread in the comC1 munity. This could limit the application of this new methodology. Anyway, this new method to classify NPF could be very useful in the near future. The results are discussed in an appropriate and balance way and the paper is well-structured. It is a significant work, concise and clearly written. I recommend publication in ACP and include some comments.**

We thank Referee #1 for their helpful suggestions. We replied to the comments below. The bold text refers to the referee's comments, and the text in italics are additions to the manuscript. The line numbers mentioned in the text below refer to the ACPD version of the manuscript.

**Specific comments:**

1. **Abstract: please, include the instrument used to obtain the data**

Instrument is added to Abstract as per suggestion from the reviewer.

2. **Line 69: "The station has accumulated 22 years. . ." Although the station is 22 years old, the dataset used in this paper is shorter, 11 years, please, indicate this here.**

We added the following to line 71:

*This study analyzes 10 years of data collected between 2006 and 2016.*

3. **Lines 116-117: "To decide whether the particle growth is observed, particle concentrations in the size range of 7 − 25 nm are examined. These particles represent the growth phase of freshly-formed clusters." Is not there any other possibility? For example, could they come from bigger particles that have suffered shrinkage? It has been observed in some sites particles below 20 nm after shrinkage.**

We thank the reviewer for interesting discussion and potential improvements of our paper. In fact, analyzing 20 years of data from Hyytiälä (Nieminen et al., 2014; Dada et al., 2017) we have not observed shrinkage of NPF in Hyytiälä. It is however a characteristic of certain events which are observed in rather urban environments (Yao et al., 2010; Alonso-Blanco et al., 2015; Salma et al., 2016).

Considering other locations for which this automated method would be applied, and for which shrinkage is observed, we can consider that the method looks for a growing mode (appearance of a peak in 7 − 25 nm) that occurs within 8 hours from the start time of the nucleating mode (Ions 2 − 4 nm). According to previous studies

reporting particle shrinkage, it takes longer than 8 hours for the particles to first grow and then shrink back to the 7–25 nm size range, so the automated method would exclude most of these cases. Also, in general, shrunk particles should not be a problem in our case as those only originate due to NPF, so we can consider then part of the growth process and still consider the whole event as NPF. Thus it should not affect our criterion.

In another situation, if there are particles unrelated to NPF (with or without shrinkage), they are too small to have originated far away, so that the main source could be related to traffic (Rönkkö et al., 2017). In the case of Hyytiälä, this source is eliminated due to very small contribution from traffic due to the semi-remote location of our measurement site. In other locations, the influence of traffic is minimized as well, since usually the rush hour occurs before the peak of NPF in the small sizes, so traffic-related are not included in the detection criteria. However, in the case an unidentified plume of pollutants (in the size range of 7 – 25 nm) occurs simultaneously or directly after the NPF peak, the automated method malfunctions. The latter type of misclassification is the major in the discrepancy between manual classification and our automated method. Accordingly, we added a clarification to the reader in the main text describing the reason behind failed statistics to section 3.5:

*Our automated method fails sometimes as the result of the simultaneous appearance of an ion burst and a pollution plume. While the misjudgment of these days as regional events is largely minimized by correcting for the background concentrations of 7–25 nm particles, erroneous classification is still possible in some cases.*

4. **Point 2.4: In order to evaluate the improvement reached with this automated method, how long time do you need to classify a year using your method? And using the manual one?**

We thank the reviewer for his suggestion to stress the importance of the automated method. Accordingly we added the following sentence to Point 2.4, Line 136.

*Once the ion and particle data have been smoothed and precipitation time stamps eliminated, classification of event takes place within a couple of minutes with a click of a button using the new automated method. This can be compared to the manual method which, for classifying one year of measurement data, would require several hours of work ant at least two people to work with it.*

5. **Lines 219-222: "The peak times of the events had the most frequent occurrence at 5 to 6 hours after sunrise, which is between 10:30 and 11:30 local time, complementing our previous assumption that NPF peaks before noon. Finally, the ending times of the events had the most frequent occurrence at 10 hours after sunrise." Do these times depend on the season? Is this possible dependency seen in the data spread shown in the figure 6?**

We thank the reviewer for his suggestion. Indeed there is a large seasonal variability affecting the start, peak and end times of regional events. Below we present the figures separated by season. Since the major fraction of the regional events occurs during spring, the combined figure presented previously in the main text reflects mostly the spring start, peak and end times. Accordingly, we replace Figure 6 in the text with the one of the spring only as it follows nicely from Figure 5. The text of section 3.4 is modified as follows.

*During spring, when most of the NPF events occur, our results (Figure 6) show that indeed RE occur after sunrise and prior to noon, with the maximum number of days occurring between the sunrise and 5 hours past sunrise. The peak times of the events had the most frequent occurrence at 5 to 6 hours after the sunrise, which is between 10:30 and 11:30 local time, complementing our previous assumption that NPF peaks before noon. Finally, the ending times of the events had the most frequent occurrence at 9 to 11 hours after sunrise. During summer the events tend to start, peak and end later than in spring, and they show lower variability in comparison to spring. This observation could be attributed to longer daylight hours and less clouds. Whereas in autumn, the events, start, peak and end earlier than in spring. Exceptionally, during winter, ion concentrations might be affected by the accumulation of snow on or around the inlets. Overall, the variability of the event start, peak and end times can be affected by the solar cycle, degree of cloudiness and seasonality.*

[Figure]

**Summer**

[Figure]

**Autumn**

[Figure]

[Figure]

Figure S1. Frequency of days at which regional events start, peak and end past sunrise. For example, most events in Spring start within 5 hours from sunrise.

6. **Lines 230-231: "For example, 65% of the originally-classified event days . . . were found to be RE, 10% were TE and 14% were IB". I don't understand how an event day obtained by the traditional method (Dal Maso et al. 2005) can be an ion burst event with C2 the new methodology. These IB events are characterized by not particle above 7 nm, so they should be considered non-event day by Dal Maso. Is there any reason for this?**

The reviewer is right, during an event day there must be a presence of nucleation mode (< 25 nm) particles for several hours. This mode is identified by an appearance of a peak after subtracting a morning background. In certain cases, when there is a high background during the morning hours, our method does not distinguish a growing mode in the size range 7-25 nm. Therefore, using our automated method, instead of identifying a day as RE, it ends up being classified as IB.

**Minor comments:**

**Check how the references are introduced in the text, sometimes not clear enough. Examples can be found in lines 35 and 48.**

Thanks to the reviewer, we modified the references in the text to a clearer format.

**Line 40: Merikanto et al., 2009 is not included in the reference list**

**Line 41: Idem for Salma et al., 2016**

**Line 111: Idem for Rose et a., 2018**

**Line 118: Idem for Yli-Juuti et al., 2011**

We revised all references used in the manuscript upon request from the reviewer.

**Lines 81-82: It is "the mobility distributions of charged and neutral aerosol particles and clusters in the size range of 0.8–47 nm and 2–42 nm, respectively, were measured with a Neutral cluster and Air Ion Spectrometer". I think it should be "the mobility distributions of neutral and charged aerosol particles and clusters. . ." to correlate with instrument list.**

We modified lines 81-82 based on the reviewer's suggestion to:

*For our proposed automated classification method, the mobility distributions of neutral and charged aerosol particles and clusters in the size ranges of 2–42 nm and 0.8–47 nm, respectively, were measured with a Neutral cluster and Air Ion Spectrometer (NAIS, Airel Ltd., Estonia, (Manninen et al., 2016; Manninen et al., 2009; Mirme and Mirme, 2013) between 2006 and 2016.*

**Line 165: "For 10 years of data (2006 – 2016). . ." Previously it has been said from 2006-2015, I think the right period is until 2016, please, unify the dates.**

Agreed, we are sorry for the typo.

**Lines 243-244: "Also, the growth can be interrupted by a sudden appearance of a cloud (Baranizadeh et al., 2014;Dada et al., 2017)." This idea already appears two lines before. Remove one of the sentences**

We thank the reviewer, and removed the sentence on Lines 243 – 244 and added the citation of Baranizadeh et al., 2014 to the previous sentence. Paragraph becomes as follows:

*The interruption mechanisms may include appearance of clouds (Baranizadeh et al., 2014; Dada et al., 2017), resulting in decreased radiation essential for particle formation and growth (Jokinen et al., 2017), or a change in the origin of arriving air masses from a clean to a rather polluted sector (Sogacheva et al., 2005).*

**Line 302: the DeSerio (2008) reference is incomplete**

Since the DeSerio(2008) refers to a report published online, we replaced the reference with a reference to a book which introduces the definition of Savitzky-Golay smoothing filters (Orfanidis, 1995).

*Orfanidis, Sophocles J. Introduction to signal processing. Prentice-Hall, Inc., 1995.*

References

Alonso-Blanco, E., Gómez-Moreno, F., Núñez, L., Pujadas, M., Cusack, M., Artíñano, B. J. A. C., and Discussions, P.: Towards a first classification of aerosol shrinkage events, 25231-25267, 2015.

Dada, L., Paasonen, P., Nieminen, T., Buenrostro Mazon, S., Kontkanen, J., Peräkylä, O., Lehtipalo, K., Hussein, T., Petäjä, T., Kerminen, V. M., Bäck, J., and Kulmala, M.: Long-term analysis of clear-sky new particle formation events and nonevents in Hyytiälä, Atmos. Chem. Phys., 17, 6227-6241, 10.5194/acp-17-6227-2017, 2017.

Manninen, H. E., Petäjä, T., Asmi, E., Riipinen, I., Nieminen, T., Mikkilä, J., Hõrrak, U., Mirme, A., Mirme, S., and Laakso, L.: Long-term field measurements of charged and neutral clusters using Neutral cluster and Air Ion Spectrometer (NAIS), Boreal Environ. Res, 14, 591-605, 2009.

Manninen, H. E., Mirme, S., Mirme, A., Petäjä, T., and Kulmala, M.: How to reliably detect molecular clusters and nucleation mode particles with Neutral cluster and Air Ion Spectrometer (NAIS), Atmospheric Measurement Techniques, 2016.

Mirme, S., and Mirme, A.: The mathematical principles and design of the NAIS–a spectrometer for the measurement of cluster ion and nanometer aerosol size distributions, Atmospheric Measurement Techniques, 6, 1061-1071, 2013.

Nieminen, T., Asmi, A., Dal Maso, M., Aalto, P. P., Keronen, P., Petäjä, T., Kulmala, M., and Kerminen, V.-M.: Trends in atmospheric new-particle formation: 16 years of observations in a boreal-forest environment, Boreal Env. Res., 19, 2014.

Orfanidis, S. J.: Introduction to signal processing, Prentice-Hall, Inc., 1995.

Rönkkö, T., Kuuluvainen, H., Karjalainen, P., Keskinen, J., Hillamo, R., Niemi, J. V., Pirjola, L., Timonen, H. J., Saarikoski, S., and Saukko, E. J. P. o. t. N. A. o. S.: Traffic is a major source of atmospheric nanocluster aerosol, 114, 7549-7554, 2017.

Salma, I., Németh, Z., Weidinger, T., Kovács, B., Kristóf, G. J. A. C., and Physics: Measurement, growth types and shrinkage of newly formed aerosol particles at an urban research platform, 16, 2016.

Yao, X., Choi, M., Lau, N., Lau, A. P., Chan, C. K., Fang, M. J. A. s., and technology: Growth and shrinkage of new particles in the atmosphere in Hong Kong, 44, 639-650, 2010.

---

## Author Comment (AC2) · 19 Nov 2018

**New particle formation has been demonstrated to play important roles in air quality and climate change. It's essential to classify the new particle formation events and non-events days accurately that can reduce the uncertainty when evaluating the contribution of NPF to aerosol and CCN budget. Previous methods were kind of subjective, and resulted in a poor comparability. This study present an automated method, which is more objective, to classify days into four categories including NPF events, non-events and two classes in between. This automated method was applied in a 10-year NAIS dataset at SMEAR II station. The classification using this methods almost matched the original method, but provided more reliable categories. Therefore, this automated method has the potential to be promoted widely. The manuscript is overall well written C1 ACPD Interactive comment Printer-friendly version Discussion paper and documented. The topic fits well in the scope of ACP. I recommend this manuscript can be published after some revisions.**

We thank Referee #2 for their helpful suggestions. We replied to the comments below. The bold text refers to the referee's comments, and the text in italics are additions to the manuscript. The line numbers mentioned in the text below refer to the ACPD version of the manuscript.

1. **A NAIS is needed to use this "new" method, which is not easy to be promoted. Can it be used with a SMPS or a DMPS? Hyde have SMPS/DMPS dataset, did the author compare the results that using a NAIS with a SMPS/DMPS? Are they identical?**

Our new method uses NAIS in ion mode to account for the process that occurs below 3 nm and represents the early steps of new particle formations which cannot measured by a typical DMPS system (Aalto et al. 2001). While the NAIS is not yet a wide spread instrument, it is important for this study since the ions play a very important role during the early stages of NPF. At these small sizes, the NAIS provides accurate ion concentrations in comparison to DMPS systems which are rather uncertain as seen in the figure below from Wiedensohler et al. (2012). We conclude that, unfortunately, our method is not applicable for data obtained merely from SMPS/DMPS measurements.

[Figure]

*Figure S 1 Comparison of particle number size distributions of different mobility particle spectrometers by Wiedensohler et al. 2012.*

2. **Line 154-156: definition of region events is "initiated over a large area including the measurement location and the particles continue to grow to bigger sizes". Since the SMEAR II station is a surface measurement site, how did the author make sure the identified "regional event" occurred over a large area?**

As discussed in a very recent review article (Kerminen et al., 2018), surface measurements at a fixed location can in principle distinguish whether the observed NPF events are regional rather than local in their character. Our classification criteria were designed so that what we call "regional events" would really represent large-scale production of new aerosol particles. The spatial extent of apparently "regional NPF" has been studied in more than 10 individual studies conducted in Europe, North America and China. The general conclusion from these studies is that the spatial extend of regional NPF is typically a few hundreds of km, possibly exceeding 1000 km in some cases (see section 3.1.3 in Kerminen et al., 2018).

3. **Transport events, is there any more evidence to support the definition? Any other possibility that other sources but not NPF contribute to the 7-25 nm particle?**

Our assumption regarding the transported events comes from the observation of downward flux of particles. One example is shown below by Leino et al. (2018). Interestingly, this is one of many similar events for which we are dedicating a complete separate study including more flight measurements above Hyytiälä (Lampilahti et al. 2019, In Prep).

However, there might for sure be contribution of a different source, for example the time period between midnight and 10 am in the figure below, and which is most possibly attributed to a pollution plume carried to our measurement location and which interferes with our automated method.

[Figure]

Figure 5. New particle formation event at SMEAR II station in Hyytiälä on 13th August 2015. Panel a) shows the number size distribution measured by Differential Mobility Particle Sizer at ground level inside the forest canopy. Start and end times of two measurement flights were marked by vertical lines in figure. Panel b shows the particle flux measured at 23 m above ground level at the station. Negative particle flux indicates particles flux downwards.

*Figure S 2 New particle formation event at SMEAR II station in Hyytiälä on 13th 550 August 2015. From Leino et al. 2018, ACPD.*

4. **Nighttime events: there are some regional events those were started and stopped before sunrise (Fig. 6)? Is it mean they are typical nighttime NPF events? Did they have the "banana" shape? If not, it means these events were not class A event, but still be defined as regional events (see comment 2)?**

We thank the reviewer for mentioning this point, we agree that there are a couple of days that have an untypical behavior in comparison to the majority of the days studied. Indeed, we do not observe typical 'bananas' that start during nighttime in Hyytiälä (Buenrostro Mazon et al., 2016; Rose et al., 2018). Our automated method matches the manual classification up to 94% and fails mostly during winter, as shown in Figure S3. The ion concentrations might be disrupted during winter due to snow accumulation. Based on the reviewer's comments we re-considered distributing the start, peak and end times analysis over the seasons, and replaced Figure 6 in the text by the similar one from only spring which is representative of the whole data set and still follows nicely from Figure 5. The results show similarity between all days and spring since the majority of RE occur during spring. Also, the redistribution of the plot into seasons confirms that these events that start and end before sunrise, are indeed in winter. The text in section 3.4 is modified as follows.

*During spring, when most of the NPF events occur, our results (Figure 6) show that indeed RE occur after sunrise and prior to noon, with the maximum number of days occurring between the sunrise and 5 hours past sunrise. The peak times of the events had the most frequent occurrence at 5 to 6 hours after the sunrise, which is between 10:30 and 11:30 local time, complementing our previous assumption that NPF peaks before noon. Finally, the ending times of the events had the most frequent occurrence at 9 to 11 hours after sunrise. During summer the events tend to start, peak and end later than in spring, and they show lower variability in comparison to spring. This observation could be attributed to longer daylight hours and less clouds. Whereas in autumn, the events, start, peak and end earlier than in spring. Exceptionally, during winter, ion concentrations might be affected by the accumulation of snow on or around the inlets. Overall, the variability of the event start, peak and end times can be affected by the solar cycle, degree of cloudiness and seasonality.*

[Figure]

**Summer**

[Figure]

**Start time of Regional Events**

**Peak time of Regional Events**

**End time of Regional Events**

Number of Days

Hours from Sunrise

**Autumn**

[Figure]

**Start time of Regional Events**

**Peak time of Regional Events**

**End time of Regional Events**

Number of Days

Hours from Sunrise

[Figure]

Figure S3. Frequency of days at which regional events start, peak and end past sunrise. For example, most events in Spring start within 5 hours from sunrise.

5. **Figure 2: it's better to give an example to show the variation of 2-4 nm particles and 7-25 nm particles in one event.**

We agree with the reviewer that showing an example of the variation of ions and particles on the same day improves the quality of our method, accordingly we changed the figures to the same day as shown below.

[Figure]

[Figure]

**References**

Buenrostro Mazon, S., Kontkanen, J., Manninen, H. E., Nieminen, T., Kerminen, V.-M., and Kulmala, M.: A long-term comparison of nighttime cluster events and daytime ion formation in a boreal forest, Boreal Env. Res., 21, 242-261, 2016.

Carnerero, C., Pérez, N., Reche, C., Ealo, M., Titos, G., Lee, H.-K., Eun, H.-R., Park, Y.-H., Dada, L., and Paasonen, P. J. A. C. P. D., https://doi. org/10./acp--173, in review: Vertical and horizontal distribution of regional new particle formation events in Madrid, 2018.

Hussein, T., Junninen, H., Tunved, P., Kristensson, A., Dal Maso, M., Riipinen, I., Aalto, P., Hansson, H.-C., Swietlicki, E., Kulmala, M. J. A. C., and Physics: Time span and spatial scale of regional new particle formation events over Finland and Southern Sweden, 9, 2009.

Kerminen, V.-M., Chen, X., Vakkari, V., Petäjä, T., Kulmala, M., and Bianchi, F. J. E. R. L.: Atmospheric new particle formation and growth: review of field observations, 13, 103003, 2018.

Leino, K., Lampilahti, J., Poutanen, P., Väänänen, R., Manninen, A., Mazon, S. B., Dada, L., Nikandrova, A., Wimmer, D., Aalto, P. P., Ahonen, L. R., Enroth, J., Kangasluoma, J., Keronen, P., Korhonen, F., Laakso, H., Matilainen, T., Siivola, E., Manninen, H. E., Lehtipalo, K., Kerminen, V.-M., Petäjä, T., and Kulmala, M.: Vertical profiles of sub-3 nm particles over the boreal forest Atmos. Chem. Phys. Discuss. (Submitted), 2018.

Németh, Z., and Salma, I.: Spatial extension of nucleating air masses in the Carpathian Basin, J Atmospheric Chemistry

Physics, 14, 8841-8848, 2014.

Rose, C., Zha, Q., Dada, L., Yan, C., Lehtipalo, K., Junninen, H., Mazon, S. B., Jokinen, T., Sarnela, N., Sipilä, M., Petäjä, T., Kerminen, V.-M., Bianchi, F., and Kulmala, M.: Observations of biogenic ion-induced cluster formation in the atmosphere, Science Advances, 4, 10.1126/sciadv.aar5218, 2018.

Salma, I., Németh, Z., Kerminen, V.-M., Aalto, P., Nieminen, T., Weidinger, T., Molnár, Á., Imre, K., Kulmala, M. J. A. C., and Physics: Regional effect on urban atmospheric nucleation, 16, 8715-8728, 2016.

Vana, M., Komsaare, K., Hõrrak, U., Mirme, S., Nieminen, T., Kontkanen, J., Manninen, H. E., Petäjä, T., Noe, S. M., and Kulmala, M. J. B. E. R.: Characteristics of new-particle formation at three SMEAR stations, 2016.

Wiedensohler, A., Birmili, W., Nowak, A., Sonntag, A., Weinhold, K., Merkel, M., Wehner, B., Tuch, T., Pfeifer, S., and Fiebig, M. J. A. M. T.: Mobility particle size spectrometers: harmonization of technical standards and data structure to facilitate high quality long-term observations of atmospheric particle number size distributions, 5, 657-685, 2012.